# Influence of the Bioactive Diet Components on the Gene Expression Regulation

**DOI:** 10.3390/nu13113673

**Published:** 2021-10-20

**Authors:** Justyna Mierziak, Kamil Kostyn, Aleksandra Boba, Magdalena Czemplik, Anna Kulma, Wioleta Wojtasik

**Affiliations:** 1Faculty of Biotechnology, University of Wrocław, Przybyszewskiego 63/77, 51-148 Wroclaw, Poland; aleksandra.boba@uwr.edu.pl (A.B.); magdalenaczemplik@tlen.pl (M.C.); anna.kulma@uwr.edu.pl (A.K.); 2Department of Genetics, Plant Breeding & Seed Production, Faculty of Life Sciences and Technology, Wroclaw University of Environmental and Life Sciences, pl. Grunwaldzki 24A, 50-363 Wroclaw, Poland; kamil.kostyn@upwr.edu.pl

**Keywords:** bioactive diet components, gene expression, DNA methylation, histone modification, non-coding RNA, signalling cascades, nuclear receptor

## Abstract

Diet bioactive components, in the concept of nutrigenetics and nutrigenomics, consist of food constituents, which can transfer information from the external environment and influence gene expression in the cell and thus the function of the whole organism. It is crucial to regard food not only as the source of energy and basic nutriments, crucial for living and organism development, but also as the factor influencing health/disease, biochemical mechanisms, and activation of biochemical pathways. Bioactive components of the diet regulate gene expression through changes in the chromatin structure (including DNA methylation and histone modification), non-coding RNA, activation of transcription factors by signalling cascades, or direct ligand binding to the nuclear receptors. Analysis of interactions between diet components and human genome structure and gene activity is a modern approach that will help to better understand these relations and will allow designing dietary guidances, which can help maintain good health.

## 1. Introduction

Application of new technologies and development of new fields of knowledge, such as nutrigenetics and nutrigenomics, allow for individual dietary recommendation. Nutrigenomics comprises investigations on interactions between nutrients and gene expression and identification of mechanisms that decide how food elements influence human health. Nutrigenetics focuses on genetic differences in genome of individual patients and on using analysis of these alterations for formulation of dietary guidance for personalized nutrition [1]. Transcriptome analysis constitutes a key tool for observation of the gene expression alteration in response to different factors. Diet, physical activity and drugs can alter gene expression and thus influence the risk of pathological changes in an organism [2,3]. Comparison of differential diet-dependent transcriptomes with the transcriptomes of healthy and sick populations enables the generation of biomarkers helpful for healthy and well-chosen diet preparation [4]. Human genetic variations within different populations results from evolutionary adaptation to different environment conditions, including food accessibility [5,6]. SNPs (single-nucleotide polymorphism) are the most common genetic variability, occurring every 500–2000 bp in the human genome [7]. The consequence of such mutations is generation of different RNA molecules and alterations in protein structure and function encoded by mutated genes, which can alter homeostasis of an organism and lead to diseases or health disorders [8,9]. SNP analysis is an important molecular tool for examining the nutrients’ impact on human health [10].

Nutrigenomics and nutrigenetics focus on three main areas, namely the interactions of diet components and genome, organism homeostasis, and personalized nutrition. Studies focused on these areas allow for better understanding of molecular interactions between the consumed food and genome, and recognizing the effects of such interactions will help in preparation of personalized diets in order to prevent or support treatments of diseases [11,12].

## 2. Mechanisms of Diet Components and Gene Expression Interaction

Bioactive diet components influence gene expression through changes in the chromatin structure (including DNA methylation, histone modification), non-coding RNA, activation of transcription factors via signalling cascades, or direct ligand binding to the nuclear receptors (Figure 1).

### 2.1. Chromatin Structure (Including DNA Methylation, Histone Modification, Telomere Length)

A significant and interesting issue within nutrigenetics and nutrigenomics is the influence of diet components on epigenetic alteration of genome. Epigenetic changes are heritable alterations of gene expression and chromatin organization, which do not result from changes in DNA sequence. Main epigenetic modifications consist of DNA methylation changes and histone modification. Food components are among factors that can trigger epigenetic changes [13], and abnormal profiles of epigenetic changes can lead to diseases. Unlike genetic changes, which are stable, epigenetic changes can be reversible, which suggests that humans can modulate them by their lifestyle and diet and even prevent the onset of diseases in their progeny [14,15].

DNA methylation is one of the best-known epigenetic modifications. It consists of attaching methyl groups to the nitrogen bases of nucleotides, mainly to cytosine and less often to adenine. DNA methylation is related to the regulation of gene expression and modulation of the chromatin structure but also participates in processes such as inactivation of the X chromosome or parent imprinting. Disturbances in the methylation profile have been found in many diseases. The methyl group is transferred from S-adenosyl-methionine (SAM) to the carbon-5 of cytosine, resulting in 5-methyl cytosine (5mC). This process takes place mainly in CG context or the so-called CpG sites. DNA methyltransferases are responsible for attaching methyl groups to DNA during de-novo replication and methylation as well as conservative methylation, related to the passing of epigenetic information to daughter cells. In humans, these are DNMT1, DNMT3A, and DNMT3B methyltransferases [16,17]. DNA methylation regulates gene expression (mainly gene silencing) by recruiting proteins involved in gene repression (the MBD proteins (methyl-CpG binding proteins)), the UHRF proteins (ubiquitin-like, containing PHD and RING finger domain protein), and the zinc-finger proteins or by hindering the binding of transcription factors to DNA [18].

The DNA methylation pattern in the genome changes as a result of both DNA methylation and demethylation. DNA demethylation can be active or passive. As DNMT1 actively maintains DNA methylation during cell replication, its inhibition or dysfunction allows newly incorporated cytosine to remain unmethylated and consequently reduces the overall methylation level. Active DNA demethylation can occur in both dividing and non-dividing cells and is driven enzymatically, based on the DNA-BER (Base Excision Repair) system with the participation of DNA glycosylases. Several mechanisms of active DNA demethylation have been proposed. One of them is 5mC deamination to thymine with the participation of AID/APOBEC proteins (activation-induced cytidine deaminase/apolipoprotein B mRNA-editing enzyme complex). Deamination of the amine to a carbonyl group by AID/APOBEC effectively converts 5mC into thymine, thus creating a G/T mismatch and inducing the BER pathway to correct the base. Another mechanism is the process mediated by the ten-eleven translocation (Tet) enzymes Tet1, Tet2, and Tet3. Tet enzymes add a hydroxyl group to the 5mC methyl group to form 5-hydroxymethylcytosine (5hmC). Two separate mechanisms (iterative oxidation by Tet enzymes and deamination by AID/APOBEC) can convert 5hmC back into cytosine. The demethylation process plays an important role in the human body, as it is necessary during the reactivation of silenced genes or incorrectly methylated bases [18]. DNA methylation profile may alter as a result of SNP, environmental factors, as well as diet components. There are three ways in which nutrients influence the methylation patterns of DNA. The first is to provide the substrates necessary for proper DNA methylation; the second, changing the activity of the enzymes regulating the one-carbon cycle, the third, providing cofactors that modulate enzymatic activity of DNMT. All the three paths are interconnected, and often, a particular dietary component works in more than one way [19].

S-Adenosylmethionine (SAM) is a methyl-donor and is synthesized in the methionine cycle from several precursors present in the diet: methionine, folate, choline, betaine, and vitamins B2, B6, and B12 (Table 1). Reduced availability of methyl donors should result in low SAM synthesis and global DNA hypomethylation. There is no simple correlation between the concentration of methyl donors and the level of DNA methylation, as other mechanisms may contribute to this. One carbon (C1) metabolism utilizes a variety of nutrients, such as glucose, vitamins, and amino acids, to fuel a variety of metabolic pathways that utilize these one-carbon units and is essential for many cellular processes, including the DNA methylation. One carbon cycle requires serine, folic acid, and methionine, which make the folic acid and methionic acid cycles highly related and essential for cellular processes. In addition, many components of the diet regulate the activity of enzymes involved in single-carbon metabolism [19]. Folic acid has attracted a great deal of attention from researchers studying the effects of dietary components on DNA methylation. This is because folate plays a key role in C1 metabolism through its conversion to *N*-5-methyltetrahydrofolate, which in turn is converted to *S*-adenosyl methionine, the CH_3_ donor in DNA methylation. Ethanol is an antagonist of folate and monocarbon metabolism. Other nutrients involved in monocarbon metabolism are vitamins B2, B6, and B12; methionine; choline; and betaine, but they show a weaker modulating effect on DNA methylation [20]. Evidence from animal, human, and in-vitro studies suggest that the epigenetic effects of folate on DNA methylation are highly complex. The effects are gene and site specific and appear to depend on the cell type, target organ, degree and duration of folate manipulations, interactions with other methyl group donors and dietary factors, and genetic variants in the folate metabolic pathways. However, now, most of the evidence suggests that high folate deficiency in the body causes global DNA hypomethylation and disturbance of the normal methylation pattern of genes involved in many important physiological processes [21]. Vegetable products that are particularly rich in folic acid include leafy vegetables, broad beans, green peas, beets and tomatoes, citrus fruits, nuts, sunflower seeds, and cereals. It is also found in animal products, mainly eggs, cheese, liver, and yeast. Folic acid supplied to the body with food may not always be fully used, either due to the conditions of storage and preparation of products or individual properties related to their absorption and metabolism; therefore, in some cases, folic acid supplementation is recommended [22]. Zinc acts as a cofactor for several enzymes in the pathway producing methyl group donor. Zinc deficiency may cause a deficit of methyl groups, and as a consequence, the expression level of certain genes may be disturbed, and the risk of disturbances in cell development may increase [23]. The increase in DNA demethylation can also be caused by vitamin C, which is connected with the increased expression of DNMT1, DNMT3a, and the mRNA expression of Tet2 and Tet3 [24].

Epigallocatechin-3-gallate (EGCG), a polyphenol member, can reduce global DNA methylation levels, DNA methyltransferase (DNMT) activity, messenger RNA (mRNA), and protein levels of DNMT1, DNMT3a, and DNMT3b. EGCG can directly inhibit the DNMT catalytic site. Catechins, which are also polyphenols, appear to inhibit DNMT activity through increasing the intracellular S-adenosylhomocysteine (SAH) levels. Quercetin, an important dietary flavonoid present in different vegetables, fruits, nuts, tea, red wine, and propolis, can also inhibit DNA methylases. Another flavonoid found in plant products, myricetin, has even stronger inhibitory potential against DNMT than quercetin. Other dietary phenolic compounds, including hesperetin, naringin, apigenin, and luteolin, can also modulate DNA methylation by indirectly regulating DNMT activity through regulating the SAM and SAH ratio [58,59,60]. Sulforaphane, an isothiocyanate found in vegetables from the Brassicaceae family, can also influence the abnormally methylated genes by modulating DNMT expression [61,62] (Table 1).

The polyphenolic extract of Annurca apples, a variety native to southern Italy, shows strong inhibition of DNMT methyltransferases, thus restoring the correct expression of silenced genes [63]. Lycopene, a carotenoid found, among other things, in tomatoes, also exhibits demethylating activity [64]. Cocoa is another nutritional product rich in polyphenols that affect DNA methylation. In-vitro experiments suggest that cocoa may exert this effect partially via the down-regulation of DNMTs, MTHFR (methylenetetrahydrofolate reductase), and MTRR (methionine synthase reductase) genes, which are key genes involved in this epigenetic process [65]. Data from animal model studies indicate that ingestion of genistein, the major phytoestrogen in soy, may induce persistent DNA hypermethylation in offspring [66].

Methyltransferases can be responsible for heritable DNA methylation changes. One of the methyltransferases, DNMT3, is responsible for DNA methylation during embryogenesis. Mother’s diet and environmental factors can influence the methylation profile during embryogenesis [67,68]. Protein restriction is a frequently used model of maternal malnutrition. Feeding pregnant rats a low-protein diet resulted in global or locus-specific changes in DNA methylation [69]. Human data also show that the offspring of mothers who have experienced a history of famine are more likely to develop metabolic diseases that are associated with epigenetic changes that have occurred in foetal life. A low-protein and low-calorie diet leads to both hypomethylation and hypermethylation at specific loci in offspring [70,71,72,73]. A high-fat, high-calorie maternal diet can also alter DNA methylation and gene expression in offspring [74,75,76]. In adulthood, dietary habits can influence the methylation pattern but to a lesser extent than in developing individuals. Both high- and low-calorie diets, especially if used for a long time, can lead to epigenetic changes in the body and then affect its health [77,78]. Research on obese patients who applied a low-calorie diet in order to reduce weight revealed significant differences in DNA methylation patterns in both those who exhibited high or low loss of weight [79]. The differences in the transcriptome of people well and poorly responsive to caloric restriction are mainly related to the genes associated with body weight control and insulin secretion [80].

Short fasting periods can influence health through DNA methylation [81]. In studies in which participants were subjected to a high-calorie diets, changes were observed in both gene expression and methylation patterns in human adipose tissue and muscle. These changes were not fully reversed by a low-calorie diet, suggesting that changes in methylation at certain loci may accumulate over time. DNA methylation of individual genes and CpG sites can be regulated differently by the supply of saturated and polyunsaturated fatty acids (Table 1). Excessive supply of saturated fatty acids leads to increased methylation of genes in adipose tissue, especially genes that are involved in the carbohydrate metabolism, lipid metabolism, and oxidative phosphorylation. It must be noticed that alteration in DNA methylation in relation to diet (e.g., rich in compounds serving as methyl donors (folate, choline, vit. B2, B6, B12)) is gene and tissue specific at different life stages. Additionally, differences in methylation are dependent on sex and genotype.

Another epigenetic process related to the chromatin structure is the histone modification. Histones are proteins that are part of the basic chromatin unit, which is the nucleosome, which consists of 147 base pairs of DNA wrapped around the octamer of histone proteins: H2A, H2B, 2xH3, and 2xH4. The chromatin also includes the H1 linker histone. Histones undergo various modifications with the participation of enzymes to be able to dynamically modulate the structure of chromatin in order to activate or silence gene expression. Histone modifications include acetylation, methylation, phosphorylation, biotinylation, and ubiquitination and mostly concern the N-terminus of histones. Most of these modifications occur at lysine, arginine, and serine residues and regulate key processes, such as transcription, replication, and repair. Histone modifications are natural processes, but alteration in their modifications can influence changes in functions of individual genes [82].

Certain patterns of histone modification determine the binding of proteins regulating the structure of chromatin and determine its state as well as the activity of genes in its area, which is called the histone code. The histone code may be changed because various modifications of histones are potentially reversible and is dynamically regulated by a group of enzymes that add or remove covalent modifications to histone proteins. Histone acetyltransferases (HATs) and histone methyltransferases (HMTs) add acetyl and methyl groups, respectively, whereas histone deacetylases (HDACs) and histone demethylases (HDMs) remove acetyl and methyl groups, respectively, from histone proteins. Histone modifications are context dependent and can have opposing effects. Research focuses mainly on histone methylation and acetylation. Methylation may involve both the activation and silencing of gene expression, whereas acetylation mainly relates to gene activation [83,84].

Nutrients can influence alteration in histone modification through interacting with histone deacetylases. Butyrate (dietary fibre fermentation), diallyl sulfide (garlic), sulforaphane (brassica sp.), curcumin, polyphenols from garlic, green tea or cinnamon, and soybean genistein belong to compounds that inhibit those enzymes. Green tea polyphenols and copper can inhibit HATs [85,86,87,88,89] (Table 1).

Resveratrol, the active compounds of red grapes, is the activator of sirtuin 1 (SIRT1), the function of which is deacetylation of histones and other proteins [90]. Sirtuin 1 plays a key role in the rearrangement of chromatin and is involved in the regulation of some of transcription factors [91].

SAM is an essential co-factor not only for DNA methyltransferases but also for histone methyltransferases. Maternal choline deficiency, which is also associated with neural tube defects and perturbed neurogenesis in the foetus, results in diminished H3K9 methylation as well as CpG methylation [92].

In addition, research on diets, such as a high-fat, a low-protein, or a caloric restriction diet, showed that extreme dietary conditions affect multiple nutrient sensing pathways and can cause global histone modification changes [83].

Nutrigenomics and nutrigenetics allowed to indicate the relationships between particular consumed products and telomere length (TL) (Table 1). The telomere consists of DNA tandem repetitions TTAGG, which together with bound proteins protect chromosome endings and get shortened with every DNA replication round and thus determines cell lifespan [93]. Telomeres enable cells to distinguish chromosome ends from double-strand breaks and thus protect chromosomes from end-to-end fusion, recombination, and degradation. Telomeres prevent the loss of genomic DNA at the ends of linear chromosomes and in turn protect their physical integrity. Literature data indicate that the catalytic subunit of telomerase, the enzyme responsible for maintaining telomeric ends, is regulated by various epigenetic modifications in its gene promoter, including histone acetylation and methylation [94].

Diet with high consumption of fruits, vegetables, healthy fatty acid, and fibre results in longer telomere fragments [95,96]. Taking into account that there is a link between oxidative stress and the abrasion of telomere, it is likely that eating foods rich in antioxidants may have important health benefits. Bioactive ingredients contributing to the maintenance of TL length are carotenoids; vitamins A, C, D, E; polyphenols; fibre; and omega-3 fatty acids. On the other hand, pro-inflammatory diets with high consumption of sugary drinks, processed meat, as well as increased amounts of saturated fatty acids were correlated with telomere shortening [97]. Current literature suggests that following a Mediterranean diet, with high consumption of antioxidants, fibre, and vegetables as well as seeds and walnuts, is associated with longer TL [98,99].

### 2.2. Non-Coding RNA (microRNA and lnc-RNA)

Diet compounds may influence the activity of non-coding RNAs (microRNA and lnc-RNA), which possess epigenetic regulatory functions. It was shown that they modulate gene expression at various levels including transcription by associating with DNA and chromatin-modifying complexes, thereby mediating alteration of the local epigenetic landscape [100].

MicroRNA (miRNA) are short 19-14 nucleotide in length fraction of non-coding RNA that constitute an essential post-transcriptional regulatory step in gene expression [101]. They work by blocking the translation or degrading the transcript (inducing exonuclease action, decapping, or deadenylating the poly (A) tail). A single miRNA can regulate several different mRNAs. In animals, miRNAs are involved in the regulation of many cellular processes, such as proliferation, differentiation, apoptosis, and also in metabolism, immune response, hormone signalling, and cell development [102]. Disorders of miRNAs can contribute to the development of various diseases, including cancer [101]. In recent years, a great deal of controversy has been raised over the possibility of penetration of plant miRNAs through the gastrointestinal (GI) barrier, where, with the participation of extracellular vesicles, such as exosomes, they enter the circulatory system in mammals [103,104,105]. In 2012, Zang et al. [106] demonstrated for the first time that dietary miR168 can cross the GI barrier and inhibit the expression of human and mouse low-density lipoprotein receptor adapter protein 1 (LDLRAP1) in liver, which resulted in lowering of LDL removal from the plasma. In 2016, the presence of plant miR159 was demonstrated in human sera, the level of which was negatively correlated with the occurrence and development of breast cancer. This was caused by influencing the transcription factor TCF7 encoding Wnt signaling transcription factor, leading to the decrease in MYC protein levels [107]. In 2020, the presence of the SIDT1 receptor responsible for the absorption of miRNA from the diet was demonstrated in gastric pit cells in the stomach of mice, and the presence of this protein in human cells was previously indicated [108,109]. Despite the increasing amount of data indicating the possibility of penetration of miRNA from the diet, this topic is still unclear and requires more extensive research [102].

Dietary food components and especially bioactive dietary components, like vitamins (vitamin D, vitamin A, vitamin E), polyphenols (resveratrol, quercetin, catechins, curcumin), fatty acids (omega-3 and omega-6), and minerals (selenium, zinc), can affect the expression of miRNA, thus regulating gene expression and the phenotype impact [110] (Table 2).

In recent years, a great deal of information has emerged about the presence of miRNA in both human and cow’s milk. Importantly, miRNA in milk is transported in extracellular vesicles, which protects the molecules against the effects of RNase and unfavourable conditions in the gastrointestinal tract. The qualitative and quantitative composition of miRNA molecules in milk is related to many factors, such as the fraction of milk, the processing method, or, in the case of human milk, the mother’s age, health, and her lifestyle [177]. For this reason, it is difficult to establish the exact profile of miRNA in human milk [178,179,180] and thus the exact effect of miRNA from milk on a child’s development. It is known, however, that milk-derived exosomal miRNAs that target DNA methyltransferase 1 (DNMT1) (miRNA-148a, miRNA-21) and DNMT3B (miRNA-148a, miRNA-29b) have been suggested to play a fundamental epigenetic role for milk-induced FOXP3 expression and Treg stabilization. These miRNAs act to inhibit DNMT and increase the stability of FOXP3 gene expression to promote immune tolerance.

lncRNAs are longer that 200 nt non-coding RNA particles. Alteration in their function can contribute to the development of many diseases, including cancers. Sulforaphane, obtained from cruciferous vegetables like broccoli, can prevent and suppress cancer formation. Research revealed that sulforaphane influences expression of cancer-associated lncRNAs. Sulforaphane-mediated alterations in lncRNA expression are correlated with genes that regulate cell cycle, signal transduction, and metabolism [181] (Table 3).

## 3. Activation of Transcription Factors by Nutrients

Organisms deploy a number of ways to maintain metabolic and energy homeostasis, including hormones and/or the nervous system. The increasing number of research papers has shown that both primary and secondary nutrients or their derivatives regulate gene expression in a hormone-independent manner by direct interaction with cellular components (membrane or nuclear receptors). The main route of activation of transcription factors (TFs) governing the expression of their target genes after extracellular impulse perception is through cascaded signal transmission, in which the preceding elements (e.g., protein kinases) change the state/structure of the succeeding ones, leading to the activation of specific TFs.

The regulation of the metabolic pathways involved in glucose homeostasis is carried out in part by the transcriptional control of the genes coding for the regulatory enzymes of those pathways. The mechanism by which carbohydrates regulate transcription of these genes, besides the transcriptional control exercised by insulin and glucagon and their signalling cascade, was finally unravelled by the purification and characterization of the carbohydrate-responsive element binding protein (ChREBP). In response to glucose and fructose, this protein forms a heterodimer with its partner Mlx and then binds and activates the transcription of target genes that contain carbohydrate response element (ChoRE) motifs (e.g., glucokinase, pyruvate kinase, ATP citrate lyase, acetyl CoA carboxylase, fatty acid synthase) [189]. This regulation plays a critical role in sugar-induced lipogenesis and glucose global homeostasis through the coordination of hepatic intermediary metabolism, carbohydrate digestion, and transport [190,191]. ChREBP transcriptional activity can be modulated by other cofactors and transcriptional factors, such as the members of nuclear receptors family hepatic nuclear factor 4 (HNF-4), LXR, FXR, or the thyroid hormone receptor (TR) [192,193]. Glucose homeostasis and body weight is regulated also by Signal Transducer and Activator of Transcription 3 (STAT3), a transcription factor activated by different cell stimuli, like leptin, grow factors, or cytokines, such as IL-5 and IL-6 [194]. In in-vivo studies, it was shown that STAT3 activation was triggered by oleic acid, leading to intensified transcription of genes regulated by this transcription factor [195]. STAT3 protein is thought to be an important factor associated with increased risk of abdominal obesity. A high dietary saturated fatty acid intake amplifies the genetic predisposition to abdominal obesity which connected with certain STAT3 genotypes [196]. In addition, high continuous activation of STAT3 gene is connected with neoplastic transformation. STAT3 is broadly hyperactivated both in cancer and non-cancerous cells within the tumour ecosystem and plays crucial roles in reducing the expression of important immune activation regulators and promoting the production of immunosuppressive factors [197]. It was shown that high-fat diet increased the risk of prostate cancer and that palmitic acid levels were strictly connected with STAT3 activation [198].

Metabolism of glucose and lipids is regulated by a family of nuclear receptors known as the peroxisome proliferator-activated receptors (PPARs). The PPARs function as lipid sensors in a way that can be activated by both dietary fatty acids (FAs) and their derivatives in the body, consequently redirecting metabolism. Three types of these receptors are known. The PPARα isoform plays a significant role in the oxidation of fatty acids and is important in body’s response to fasting. PPARγ is abundant in the brown adipose tissue and is an important regulator of fat cells [199] and is crucial in regulating adipogenesis (through C/EBP cascade), thus playing a significant role in maintaining glucose and lipid metabolism balance. Among the identified functions of PPARδ is that of its connection with the catabolism of fatty acids and energy homeostasis [200]. It is suggested that PPARα may be the most important isoform in regulating de-novo fatty acid synthesis from carbohydrates and lipid deposition. It was shown that FASKOL (fatty acid synthase knockout in liver) mice, when fed a diet deprived of fat, were characterized by hypoglycaemia and hypercholesterolemia. This effect was reversed by PPARα agonists, which further led to the conclusion that only dietary fat or fat synthesized de novo due to fatty acid synthase activity is capable of PPARα activation, which in turn leads to gluconeogenesis [201]. In contrast, PPARδ may work as a widespread regulator in fat burning and probably could be used as a potential target in the treatment of obesity and related disorders [202]. It was proposed that the effect of fatty acids on gene expression mainly depends on the number of double bonds and the length of carbon tail, and hence, abnormal lipid profile can lead to severe aberration in cell function (Table 4). After PPAR binding, polyunsaturated fatty acids lower the level of triglycerides and increase blood HDL cholesterol fraction [203]. Fatty acids can also regulate liver X receptor (LXR), retinoid X receptor (RXR), and sterol receptor SREBP (sterol regulatory element-binding proteins) [204]. It is already known that LXRs, after forming a heterodimer with RXRs, regulate the nutrient metabolism pathways through their interactions with specific, naturally occurring oxysterols. It was found that LXRs could also form heterodimers with all the three PPAR subtypes with different binding affinities, and such receptor/receptor interactions could be regulated by ligand binding [205]. Formation of such heterodimers changes their target genes. By gel shift and in-vitro protein/protein binding assays, it has been discovered that the interactions between LXRs and PPARα are involved in fatty acid degradation, which is a reverse of the fatty acid synthesis function of PPARα [206]. Sterol regulatory element binding proteins (SREBPs) are membrane-bound transcription factors of the basic-helix–loop–helix–leucine zipper (bHLH-Zip) family that have been shown to regulate enzymes responsible for the synthesis of cholesterol, fatty acids, and the low-density lipoprotein (LDL) receptor. The target genes involved in cholesterol metabolism include the LDL receptor, 3-hydroxy-3-methylglutaryl-CoA (HMG-CoA) synthase, HMG-CoA reductase, farnesyl-diphosphate (FPP) synthase, and squalene synthase, while genes involved in fatty acid and triglyceride synthesis that are regulated by SREBPs include acetyl-CoA carboxylase (ACC), fatty acid synthase (FAS), and glycerol-3-phosphate acyltransferase [207]. Diets rich in sterols prevent proteolytic cleavage of SREBPs that facilitates its translocation to the cell nucleus, thus weakening the transcription ratio of target genes. In studies with transgenic mice overexpressing the nuclear form of SREBP-1c in the liver, 2- to 4-fold increases in mRNAs for genes involved in fatty acid synthesis were observed [208]. SREBP-1c transcription can also be induced by the activation of liver X receptor (LXR)α, a hormone nuclear receptor that is activated by oxysterols (cholesterol derivatives). It leads to the induction of expression of a number of genes connected with cholesterol removal, which share the same LXR response element, 5′-AGGTCANNNNAGGTCA-3′ [209]. It is believed that LXRα acts as a cholesterol sensor, and after SREBP-1c induction, fatty acids are produced in order to esterify cholesterol, which in turn balances the cholesterol amount in the organism [210].

Beside carbohydrates and fats, proteins are the third most important class of macromolecules that need to be received with diet. Although higher organisms are able to synthetize some amino acids, there is a group of these compounds, called essential amino acids, that must be obligatory acquired from food. Moreover, the synthesis of the remaining amino acids usually does not meet the requirement of an organism; therefore, a healthy and balanced diet must cover all the requirements in amino acids and should include proteins from different sources and in different proportions. The pool of amino acids in the body results from two counteracting processes, protein synthesis and protein breakdown. In the situation when amino acid level decreases (especially of the essential amino acids), the so-called amino acid response is triggered [325]. It leads to deacetylation of corresponding tRNAs, which in turn activates the general control nonderepressible 2 (GCN2) kinase. Then the kinase activates the eukaryotic initiation factor 2 alpha (eIF2α) [326], and the integrated stress response begins [327], in which mRNA synthesis is reduced or halted because of decreased eIF2B protein complex activity [328]. In contrast, activation of the activating transcription factor 4 (ATF4) occurs that activates of specific genes involved in the adaptation to starving situation by binding to CCAAT enhancer-binding protein (C/EBP)-ATF response elements (CARE) [329]. Under continued stress of sufficient magnitude, ATF4-induced apoptosis can also occur [330]. Activation of the AAR regulates gene expression at many steps, including chromatin structure, transcription start site, transcription rates, mRNA splicing, RNA export, RNA turnover, and translation initiation. Although major, this signaling pathway is not the only one activated during amino acid deficit. For instance, under asparagine but also leucine, isoleucine, and glutamine starvation the level of asparagine synthetase mRNA increases. A region of promoter sequence 5′-TGATGAAAC-3′ −68 nt to −60 nt was identified as amino acid response element (AARE) [331]. Noteworthy, glucose depletion does the same through activation of endoplasmic reticulum (ER) stress response pathway. Both pathways act by binding to nutrient-sensing response elements 1 and 2 (NSRE-1 and NSRE-2), thereby increasing asparagine synthase transcription [332].

Many food components can modify the neoplastic progression. The modification of metabolism of carcinogens is one of the main possible mechanisms by which food components can minimize the risk of cancer. The responses to dietary compounds that have a role in preventing cancer may be related to the diversity of the enzymes being processed and modified. Key points in the cell cycle are regulated by different protein kinase complexes that are composed of cyclin and cyclin-dependent kinase molecules. Additionally, these cell cycle key points are affected by combined dietary components. It has been proven that the dietary factors either essential or nonessential can adjust and modify the cell cycle checkpoints and, consequently, have a role in reducing the progression and proliferation of tumour [333]. Beside affecting the function of proteins essential in the process of carcinogenesis, dietary components may accelerate cell death and enhance apoptosis. Bioactive diet components, such as quercetin, curcumin, and sulphoraphane, can influence signaling pathways by inhibiting NF-κB transcription factor [334] (Table 4). Bioactive diet components can block one or more stages of NF-κB signaling pathway, such as signaling cascade, NF-κB translocation, and its interaction with DNA [329]. In xenografted tumours, curcumin upregulated the expression of TRAIL-R1/DR4, TRAIL-R2/DR5, Bax, Bak, p21/WAF1, and p27/KIP1 and inhibited the activation of NF-κB and its gene products [335]. Pretreatment with a noncytotoxic concentration of luteolin significantly sensitized both TRAIL-sensitive as well as TRAIL-resistant cancer cells to TRAIL-induced apoptosis [336]. Curcumin and also other curcuminoids form ginger regulate NF-κB transcription factor and gene products, such as COX-2 (cyclooxygenase-2) and cyclins. Curcumin inhibits mediators of NF-κB activation. Guggulsterone, one of the plant sterols, inhibits activation of NF-κB via direct interactions with kinases of this pathway [337,338]. Polyunsaturated fatty acids influence expression of genes, encoding factors that take part in inflammation processes via activation of NF-κB factor [339]. Resveratrol suppresses phosphorylation and translocation of one of the NF-κB subunits. It also blocks activation of NF-κB pathway by cancerogenic compounds. Other compounds that inhibit phosphorylation on this pathway are isoanthocyanins derived from cruciferous vegetables [340]. Green tea polyphenols inhibit activation of NF-κB via inhibition of kinase activity on this pathway [341]. Moreover, epicatechins are known to also target AP-1 and Nrf2 redox-sensitive transcription factor associated with cell proliferation, survival, differentiation, apoptosis, and stress responses [342] (Table 4). Epicatechin transiently activated the NF-κB cascade and Nrf2 signaling by stimulating PI3K/AKT and ERK pathways and induced a sustained enhancement of AP-1-binding-activity by up-regulating the nuclear levels of c-Jun. The activation of the AP-1 signaling pathway controls cell proliferation through cell-cycle regulation in hepatocytes and other cell types [343]. Numerous reports seem to be contradictory, showing epicatechin to be unable to influence AP-1 [344] and to activate AP-1 [345]. Such variation in the effects of AP-1 activity modulation may result from the specific modulation of transcription factors in different cell types and from the concentration of epicatechins. Isoanthocyanins derived from cruciferous vegetables influence the activity of Nrf2 transcription factor, which binds to the ARE responsive element and activate transcription of genes coding for enzymes, such as quinone reductase or glutathione S-transferase 2 (enzymes of phase II of detoxication). Active compounds of cruciferous vegetables affect protein complex dissociation with Nrf2 and also phosphorylation of constituents of this complex [346].

Apoptosis can be induced due to the action of p53 tumour suppressor protein. Its expression is also activated by dietary components, like trans-resveratrol [347], silibinin [348], or curcumin. These compounds lead to apoptosis of tumour cells by inhibiting anti-apoptoting proteins, like survivin or Bcl-2 [349] (Table 4).

## 4. The Influence of Bioactive Diet Components on Diseases

Epidemiological studies revealed that populations whose diet is rich in fruits and vegetables rarely get cancer diseases [350,351,352,353]. Fruits and vegetables are the great source of fibre, vitamins, and minerals but also contain such compounds as terpenes, alkaloids, and phenolics, which comprise health benefits. Over 500 compounds derived from food were identified as putative modifiers of cancerogenesis. They not only consist of plant-derived compounds but also of animal and fungal origin and metabolites derived from processes driven by human microflora [354]. These compounds regulate gene expression through changes in the chromatin structure, epigenetic changes, or activation of transcription factors.

One of the more serious problems resulting from inadequate diet selection are allergies. Although the very mechanism of allergic reactions to nutrients is relatively well studied, some aspects require detailed research, especially when they relate to epigenetic regulation. For example, it is fact that a mother’s diet (during pregnancy and breastfeeding) has an immense and lasting impact on the development of the immune system of the offspring and consequently on the occurrence of allergies. It has been shown that allergic disease intensification corresponds with the activity of PKCζ (Protein Kinase C) gene in T cells during neonatal life. It was observed that dietary supplementation of pregnant women with *ω*-3 poly-unsaturated fatty acids (PUFA)-rich fish oil leads to reduced incidence of allergies in their progeny due to elevated expression of the PKCζ gene probably regulated epigenetically through changes in DNA methylation or histone acetylation [355,356,357,358]. Fish oil supplementation affects epigenetic changes also in other genes connected with the immune system, like IL13 or T-box 21 [359]. In another study, pregnant women’s diet supplementation with olive oil was shown to influence histone acetylation in genes of immune system regulating proteins during foetal life. For example, increased acetylation of H3 histone in the FOXP3 gene (encoding one of the regulators of immune homeostasis) promoter was observed. FOXP3, as the major transcription factor in the regulatory T cells (Tregs), plays a crucial role in the induction of tolerance towards self- and environmental (including food) antigens [360]. However, it must be noted that PUFAs can be allergenic because they promote pro-inflammatory processes and thus affect DNA methylation and histone acetylation and modulate the expression of regulatory RNA forms (miRNAs, lncRNAs), and only well balanced and adequately selected diet composition must be considered.

Adequate, healthy nutrition (a diet rich in vegetables, fruits, vegetable fats, fish, eggs) is important at every stage of life because it is a factor in reducing the occurrence of not only cancer but also other non-infectious diseases, such as obesity, diabetes mellitus type 2, cardiovascular diseases, neurodegenerative diseases, or allergic diseases. Bioactive components of the diet, such as vitamins, minerals, polyphenols, carotenoids, and isothiocyanates, affect the expression of a number of genes, showing a positive effect on the occurrence and development of diseases. This happens through mechanisms such as DNA methylation, histone modifications, telomerase inhibition, and the regulating effect of RNA or transcription factors. The mechanisms of bioactive diet components together with the disorders they are active against are presented in detail in Table 1, Table 2, Table 3 and Table 4, where also literature references can be found to facilitate extended search on particular issue (it must be noted that literature search was not systematic).

## 5. Application of Bioactive Diet Components in Dietician’s Work

It is estimated that the human genome consists of over 30,000 coding genes, which generate around 100,000 functional proteins. Understanding the interactions between gene products and bioactive diet component consumption has a fundamental meaning for identification of these compounds, which will bring the highest benefits for health and will be correlated with risk of disease onset. Application of new, innovative technologies, such as microarrays, RNA interference, and nanotechnologies, provide information for identifying molecular mechanisms of bioactive components activity. Such knowledge allows proper diet application for individual phenotypes. Some general dietary recommendation adapted to specific diseases or applied in prevention will not bring the expected effects due to the individual genetic and epigenetic diversification. Analysis of association between food and gene expression allows to formulate the proper diet, which will prevent disease or bring back organism homeostasis [1,4,11,361].

Bioactive diet components, such as polyphenols, vitamins, flavonoids, carotenoids glucosinolates, isothiocyanates, terpenes, and fatty acids, are substances that can influence gene expression by transcription factors, epigenetic modification, and enzymes, which modify chromatin structure [4]. However, consuming foods with high amounts of products of selected group may sometimes be detrimental, especially if the diet is long term. For example, carotenoids belong to the most efficient physical quenchers of singlet oxygen (^1^O_2_), the deactivation of which is based on the conversion of an excess of energy to heat via the carotenoid lowest excited triplet state (^3^Carotenoid*) [362].

Conceivable damaging effects of excited carotenoids can be ignored because of their low energy and short lifetimes. However, carotenoids can also be chemical quenchers of ^1^O_2_, undergoing modifications such as oxidation or oxygenation [363]. Free radical scavenging can lead to the formation of carotenoid radical cations or anions as well as neutral carotenoid radicals. The newly formed carotenoid radical products can undergo further transformations, leading to a variety of secondary carotenoid derivatives of different reactivity. This is especially important, as the newly generated carotenoid species may no longer act as efficient antioxidants but turn into potentially harmful, pro-oxidant agents and lead to alterations in amino acids or nucleotides that result in irreversible structural modifications of proteins or nucleic acids. Results obtained from trials with high carotenoid content diets involving heavy-cigarette-smoking men indicated a significantly higher occurrence of lung cancer and total mortality in comparison to individuals obtaining the placebo [364,365].

The use of foods that can modify the epigenome translates into a dietary regimen known as the “epigenetic diet”. Such a diet may be used therapeutically for health or prophylactic purposes. Epigenetic therapy is a new area in the development of nutraceuticals, the lack of toxicity of which can be an important asset in disease prevention strategies. Recent advances in understanding the mechanisms of nutrigenomics, nutrigenetics, and nutraceuticals have led to the identification of foods capable of favourably conditioning gene expression. The epigenetic diet must include fruit and vegetables, in particular cruciferous vegetables, bean vegetables, grapes, citrus fruits, *Curcuma longa* L, garlic, as well as tea, nuts, and whole grain cereal products [366].

### Limitations to the Studies on the Influence of the Bioactive Diet Components on the Gene Expression Regulation

The data presented in this review are based mainly on human cell cultures and as such are subject to certain limitations. First of all, it is an artificial system, and the cultured cells have no chance to respond to a current factor in the way as they do in an organism, where they are exposed to a whole gamut of factors interacting to bring about any response. Moreover, treatments with particular compounds may not reflect the actual situation, where they are processed by the organism to yield a number of possible products/derivatives, which may have a quite different impact on the cell [367]. Moreover, often, there are technical discrepancies and heterogeneity in the approach to the conducted research, like different procedures of acquisition and processing as well as different conditions of storage and transport of the samples. The investigation methods are not unified, which translates to divergent extraction/isolation procedures (of RNA, DNA, proteins) and use of different reference standards and different enzyme assay protocols.

As a matter of fact, even studies on whole organisms are not free of limitations. It results from the fact that although such studies are often carried out on a representative group of subjects, they rarely take into account the inter-individual variability in the metabolism and bioavailability of the bioactive compounds nor the individual response to them [368]. The most reliable data comes from randomized controlled trials; however, when it comes to nutrition, not all of it can be ethically evaluated in this way. Therefore, even in human studies, the data mostly come from observational evidence, in particular cohort studies, where even differences in absorption in the gut are usually not evaluated, and the studies quite often lack appropriate control groups [369].

Considering the above information, in order to obtain reliable data that will undoubtedly prove the influence of a given bioactive component of the diet on the expression of selected genes, there is a need to apply standardized procedures and follow well-established methodological guidelines as well as to conduct bioavailability studies of such a component taking into account the population variability of the test and reference groups to achieve more reliable results in future research [370].

## 6. Conclusions

Bioactive diet components influence gene expression via different mechanisms, mainly by chromatin structure alteration, non-coding RNA, activation of transcription factors by signaling cascades, or direct ligand binding to the nuclear receptor. Identification of these compounds and elucidating their mechanism of function will allow more effective diet recommendation for whole population types or for individuals. Bioactive diet components play an important role in prevention and therapy of many diet-depending diseases, such as cancers, circulatory system diseases, diabetes, and obesity. It seems important to develop further the knowledge on nutrigenetics and nutrigenomics and to encourage dieticians to use this knowledge for more effective dietary recommendations.

## Figures and Tables

**Figure 1 nutrients-13-03673-f001:**
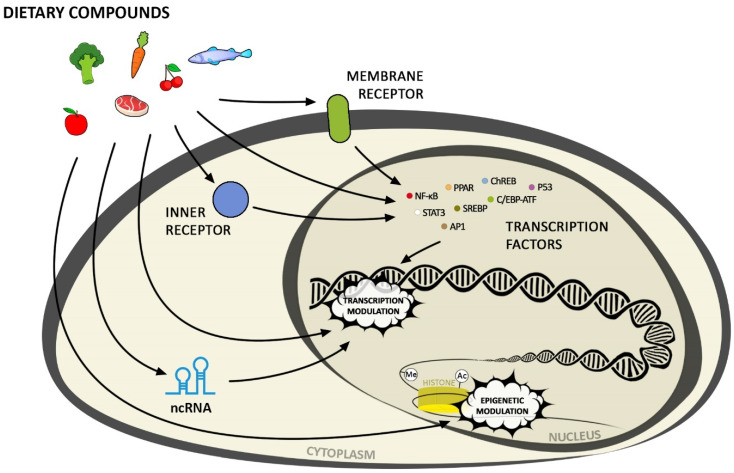
Methods of dietary compound influence on gene expression (for epigenetic modifications -(Me) stands for methyl group in methylated DNA; -(Ac) stands for acetyl group in acetylated histones).

**Table 1 nutrients-13-03673-t001:** Bioactive components of the diet and their role in epigenetic changes in the genome.

Mechanisms	Bioactive Component	Disorders	References
upregulation of DNMT	omega-3 fatty acids: DHA, EPA	colorectal cancer	[25,26]
	folic acid	colorectal cancer, breast cancer	[27,28]
	methionine	lung cancer	[29]
	vitamin A	congenital heart defects	[30]
DNA methyltransferase inhibition	kaempferol	bladder cancer	[31]
	gallic acid	lung cancer and oral cancer	[32]
	epigallocatechin-3-gallate	breast cancer, diabetic kidney disease	[33,34]
	β-caroten	colorectal cancer	[35]
	sulforaphane	breast cancer, cardiomyopathy	[36,37]
	omega-3 fatty acids: EPA	hepatocarcinoma	[38]
	vitamin A	congenital heart defects	[30]
histone deacetylase inhibition	resveratrol	breast cancer, renal cell carcinoma, colorectal cancer	[39,40,41,42]
	apigenin	prostate cancer, lung cancer	[43,44]
	luteolin	lung cancer, leukemia	[45,46]
	chrysin	melanoma	[47]
	cinnamic acid derivatives	colon and cervical cancer	[48]
	gallic acid	prostate cancer, cardiovascular diseases	[49]
	epigallocatechin-3-gallate	cardiac diastolic dysfunction, prostate cancer, acute promyelocytic leukemia	[50,51,52]
	sulforaphane	Alzheimer’s disease, melanoma, colon cancer, cardiomyopathy	[37,53,54,55]
	omega-3 fatty acids: EPA	hepatocarcinoma	[38]
	vitamin D	breast cancer	[56]
telomerase inhibition, telomere shortening	epigallocatechin-3-gallate	glioblastoma	[57]

**Table 2 nutrients-13-03673-t002:** Bioactive components of the diet and their role in miRNA level changes.

Mechanisms	Bioactive Component	Disorders	References
*↓ miR-143* and *miR-124*	curcumin	osteoarthritis	[111]
*↑ miR-99a*		retinoblastoma	[112]
*↑ miR-34a*, *miR-503*, *miR-424*	resveratrol	breast cancer	[113]
*↑ miRNA* *-* *200*		pancreatic ductal adenocarcinoma	[114]
*↑ miR-122-5p*		breast cancer	[115]
*↑ miR-200c*		colorectal cancer	[116]
*↓ miR-155*, *miR-34a* ↑ *miR-21*, *miR-181*, *miR-186*		type 2 diabetes, hypertensive patients with coronary artery disease	[117]
*↓ miR-221*		melanoma	[118]
*↑ miR-29b*	quercetin	diabetic retinopathy	[119]
*↑ miR-146a*		breast cancer	[120]
*↓ miR-206*		osteoporosis	[121]
*↓ miR-21*		breast cancer	[122]
*↓ miR-22*		oral lichen planus	[123]
*↓ miR-216a*		peripheral arterial disease	[124]
*↓ miR-21*		hepatic steatosis and fibrosis	[125]
*↓ miR-15a* and *miR-16*		hepatocellular carcinoma	[126]
*↓ miR-16*		oral cancer	[127]
*↑ hsa-miR-24*, *hsa-miR-6769b-3p*, *hsa-miR-6836-3p*, *hsa-miR-199a-3p*, *hsa-miR-663a*, *hsa-miR-4739*, *hsa-miR-6892-3p*, *hsa-miR-7107-5p*, *hsa-miR-1273g-3p*, *hsa-miR-1343*, and *hsa-miR-6089*; *↓ hsa-miR-181a-5p* and *hsa-miR-148a-3p*	apigenin	hepatocellular carcinoma	[128]
*↑ miR-34a-5p*		lung cancer	[129]
*↑ miR-152-5p*		cervical carcinoma	[130]
*↑ miRNA-215-5p*		colorectal cancer	[131]
*↑ miR-34a-5p*	luteolin	lung cancer	[132]
*↑ miR-203*		breast cancer	[133]
*↑ miR-6809-5p*		hepatocellular carcinoma	[134]
*↑ miRNA-34a*		gastric cancer	[135]
*↓ miR-21* and *↑ miR-16* and *-34a*		breast cancer	[136]
*↓ microRNA-132*		Bronchopneumonia	[137]
*↓ miRNA-301-3p*		pancreatic cancer	[138]
*↑ microRNA-340*	kaempferol	lung cancer	[139]
*↑ miR-339-5p*		colon cancer	[140]
*↓ microRNA-21*		liver cancer	[141]
*↓ miR-146a*		osteoarthritis	[142]
*↑ miR-203*		hypertension	[143]
*↑* miR-132 and miR-502c	chrysin	breast cancer	[144]
*↑* miR-9 and Let-7		gastric cancer	[145]
↓ miR-18a, miR-21, and miR-221 genes		gastric cancer	[146]
↓ microRNA (miR)-92a		atherosclerosis	[147]
*↓ miR-636*	caffeic acid	diabetic nephropathy	[148]
*↑ miR-221*	epigallocatechin-3-gallate	hepatic fibrosis	[149]
*↑ miR-548m*		hepatitis C	[150]
*↑ microRNA-let-7b*		melanoma	[151]
*↑ miR-520a-3p*		prostate cancer	[152]
*↑ miR-384*		ischemic heart disease	[153]
*↓ miR-25*		breast cancer	[154]
*↑ miR-9-3*	sulforaphane	lung cancer	[155]
*↑ miR135b-5p*		pancreatic cancer	[156]
*↓ miRNA-423-5p*		liver fibrosis	[157]
*↓ miR30a-3p*		pancreatic cancer	[158]
*↓ miR-155*		acute myeloid leukemia	[159]
*↓ miR-21*		colon cancer	
*↑ miRNA-124-3p*		nasopharyngeal cancer	[160]
*↓ miR-23b*, *miR-92b*, *miR-381,* and *miR-382*		breast cancer	[36]
*↑ miR-29a-3p* and *miR-200a*	Carotenoids: lycopene, β-carotene, lutein, astaxanthin	colorectal cancer	[161]
*↑ miR-320d*, *miR-1246* and *miRNA-1290*		neuroblastoma	[162]
*↑ miR* *-* *let* *-* *7f* *-* *1*		prostate cancer	[163]
*↑ miR-99a*	omega-3 fatty acids: DHA, EPA	breast cancer	[164]
*↑ miR-138-5p*		lung cancer	[165]
*↑ miR-34a*		multiple myeloma	[166]
*↑ miR-194* and *↓ miR-106b*		breast cancer	[167]
*↓ miR-21*		breast cancer	[168]
*↓ microRNA* *-* *155*		carotid restenosis	[169]
*↓ microRNA-20a*		gastric cancer	[170]
*↓ miR-21*		Parkinson’s disease	[171]
*↓ miRNA-146a* and *-155*	vitamin D	obesity	[172]
*↑ miR-100* and *-125b*		prostate cancer	[173]
*↑ miR-10a*	vitamin E	breast cancer	[174]
*↑ miR**-**374*, *miR**-**16*, *miR**-**199a**-**5p*, *miR**-**195,* and *miR**-**30e; ↓ miR**-**3571*, *miR**-**675,* and *miR**-**450a*	selenium	cardiac dysfunction	[175]
*↓ miR-21*, *miR-31*, and *miR-223; ↓ miR-375*	zinc	esophageal cancer	[176]

**Table 3 nutrients-13-03673-t003:** Bioactive components of the diet and their role in lncRNA level changes.

Mechanisms	Bioactive Component	Disorders	References
*↓ BRAF-activated long noncoding RNA (BANCR)*	luteolin	thyroid carcinoma	[182]
*long non-coding RNA*	epigallocatechin-3-gallate	lung cancer	[183]
*↓ lnc RNA LINC00511*		gastric cancer	[184]
*↓ lncRNAs H19*	sulforaphane	pancreatic ductal adenocarcinoma	[185]
*↓ lncRNA LUCAT1*	vitamin D	oral squamous cell carcinoma	[186]
*↓ lncRNA CCAT2*		ovarian cancer	[187]
*↑ lncRNA MEG3*		colorectal cancer	[188]

**Table 4 nutrients-13-03673-t004:** Bioactive components of the diet and their role in transcription factor activity changes.

Mechanisms	Bioactive Component	Disorders	References
PPAR activation	resveratrol	autism spectrum disorder, obesity and insulin resistance	[211,212]
	kaempferol	hyperlipidemia	[213]
	gallic acid and p-coumaric acid	type 2 diabetes	[214]
	epigallocatechin-3-gallate	Alzheimer’s disease	[215]
	lycopene	liver and lung cancer	[216]
	omega-3 fatty acids: DHA	pancreatic acinar, breast cancer, Parkinson’s disease	[171,217,218]
	folic acid	non-alcoholic steatohepatitis	[219]
	vitamin D	cerebral ischemia, metabolic syndrome	[220,221]
downregulation of PPARγ	epigallocatechin-3-gallate	obesity	[222]
NF-κB activation	quercetin	melanoma	[223]
	apigenin	multiple myeloma	[224]
	vitamin A	lung cancer	[225]
NF-κB inhibition	curcumin	gastric cancer, breast cancer, acute lung injury, oral cancer, cerebral ischemia/reperfusion (I/R) injury	[226,227,228,229,230]
	resveratrol	lung cancer, melanoma	[118,231]
	quercetin	coronary artery disease, coronary heart disease, alcohol-induced liver injury	[232,233,234]
	apigenin	colon cancer, bladder cancer, breast cancer, inflammatory bowel disease (IBD) and colitis-associated cancer (CAC)	[235,236,237,238]
	keampferol	spinal cord injury, hypertension	[143,239]
	chrysin	melanoma	[240]
	caffeic acid phenethyl ester	nasopharyngeal carcinoma, calcific aortic valve disease, periodontal diseases, glaucoma, neuropathic pain, ovarian cancer	[241,242,243,244,245,246]
	caffeic acid	hyperglycemia	[247]
	epigallocatechin-3-gallate	temporal lobe epilepsy, lung cancer	[248,249]
	sulforaphane	prostate cancer	[250]
	lycopene	pancreatic cancer, prostate and breast cancer	[251,252]
	omega-3 fatty acids: DHA	liver cirrhosis, breast cancer, pancreatic cancer	[218,253,254]
	folic acid	steatohepatitis	[255]
	selenium	prostate cancer, breast cancer, type 2 diabetes	[256,257,258,259]
	vitamin D	obesity	[172]
	vitamin E	prostate cancer	[260]
Nrf2 activation	curcumin	cerebral ischemia/reperfusion (I/R) injury	[230]
	resveratrol	diabetic cardiomyopathy	[261]
	apigenin	vitiligo, diabetic nephropathy	[262,263]
	luteolin	colon cancer, colorectal cancer, diabetic cardiomyopathy	[264,265,266]
	epigallocatechin-3-gallate	hyperglycemia, obesity, colorectal cancer, retinal ischemia-reperfusion	[267,268,269]
	sulforaphane	colon cancer, Alzheimer’s disease, cardiomyopathy	[37,270,271]
	omega-3 fatty acids: DHA	traumatic brain injury	[272]
	vitamin A	cholestasis	[273]
	vitamin E	chronic liver injury	[274]
Nrf2 inhibition	apigenin	lung cancer	[275]
	luteolin	colon cancer	[276]
	keampferol	non-small cell lung cancer	[277]
	chrysin	breast cancer, glioblastoma	[278,279]
	gallic acid	psoriasis-like skin disease, respiratory diseases	[280,281]
	vitamin E	asthma	[282]
	zinc	diabetic nephropathy	[283]
AP-1 inhibition	curcumin	renal cell carcinoma, bladder cancer, oral cancer	[229,284,285]
	gallic acid	nasopharyngeal cancer	[286]
	quercetin	coronary heart disease	[233]
	apigenin	bladder cancer	[236]
	vitamin E	leukemia	[287]
	zinc	prostate cancer	[288]
STAT3 inhibition	curcumin	osteosarcoma, myeloproliferative neoplasms, retinoblastoma	[112,289,290]
	resveratrol	osteosarcoma, colon cancer, ovarian cancer, cervical cancer	[291,292,293,294]
	quercetin	hepatocellular carcinoma, alcohol-induced liver injury	[234,295]
	apigenin	hepatocellular carcinoma, breast cancer, colon cancer, visceral obesity, inflammatory bowel disease (IBD) and colitis-associated cancer (CAC)	[237,238,296,297,298,299]
	luteolin	gastric cancer, pancreatic cancer, hepatic fibrosis, lung adenocarcinoma	[300,301,302,303]
	keampferol	diabetic nephropathy	[304]
	chrysin	bladder cancer	[305]
	gallic acid	non-small cell lung cancer	[306]
	omega-3 fatty acids: DHA	renal cancer, multiple myeloma, pancreatic cancer	[254,307,308]
	sulforaphane	nasopharyngeal cancer, glioblastoma multiforme	[160,309]
activation of p53	curcumin	gastric cancer, neuroblastoma, renal cell carcinoma	[310,311,312]
	resveratrol	prostate cancer, colon cancer, hepatocellular carcinoma, glioblastoma multiform, neuroblastoma, thyroid cancer	[313,314,315,316,317,318,319]
	epigallocatechin-3-gallate	liver cancer	[320]
	vitamin D	endometrial cancer	[321]
inhibition of p53	resveratrol	osteoporosis, breast cancer	[322,323]
	vitamin E	breast cancer	[324]

## Data Availability

Not applicable.

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
