# Peer review of "Influence of the Bioactive Diet Components on the Gene Expression Regulation"

_nutrients, 2021, doi:10.3390/nu13113673_

Round 1

Reviewer 1 Report

The article is well structured, comprehensive, and provides a wide perspective on nutrigenetics and nutrigenomics, but the article has formatting mistakes that should be considered.

Line 57: Figure 1 is abbreviated in the text Line 110 – 114: The text size is too large Line 116 – 118: The text size is too large Line 174 – 176: The text size is too large Line 180 – 182: The text size is too large Line 198 – 199: The text size is too large Line 211 – 213: The text size is too large Line 252 – 255: The text size is too large Line 257 – 266: The text size is too large Line 484: The text size is too large

Author Response

Thank you for the revision and comments to our manuscript.
We changed the “Fig.” abbreviation to “Figure”. As for the font size in some parts of the text, we checked the initial version of our manuscript (the one we uploaded to the editorial system) and there were no such formatting mistakes compared to the version that can be downloaded from the MDPI website, thus we suspect that the changes might have occurred due to some file transformation by the submission system. We will inform the Editor about the problem and hope it will not happen again.

Reviewer 2 Report

With real interest, I read the manuscript nutrients-1396275. The issue this work addresses is worth investigation and becomes more and more important.

1. A huge, I would even say impressive, work is hidden behind this manuscript, especially Table 1. However:

1A. Was the literature search systematic? If not, this should be mentioned as a limitation as some important findings might have been omitted.

1B. I would divide Table 1 into several smaller subtables, e.g. with classical epigenetic mechanisms, miRNA, transcription factors.

1C. The content of this table needs to be, at least concisely, discussed in the main text of the manuscript, the best in sections corresponding to the sub-tables mentioned above.

2. In continuation, some topics must be naturally ignored as not all possible aspects can be covered. But I would definitely not omit allergies. There are reports on the role of food compounds such as fish or olive oil-deriving compounds such as PUFA or polyphenols in regulating the mechanisms underlying the development of allergies (PMID: 28159873 and 30823645). Or milk components such as fat-soluble vitamins A and D, as well as the water-soluble B vitamins, and PUFA (PMID: 28933365, 33193294). And others. Those should be mentioned.

3. The Authors discuss indirect effects of dietary compunds on gene expression. But there are also direct effects exerted by regulatory RNAs present in foods such as:

3A. milk (PMID: 26707195, 34067156).

3B. and others (PMID: 33567805, 33668787).

Those should be mentioned, at least shortly.

4. I guess, SCFA as indirect dietary compunds (gut bacteria oligosaccharide fermantation metabolites) can be ignored in this work (as they currently are).

5. Please, check the list of abbreviations. Some of them are never used in the manuscript, e.g. PUFA.

6. Abbreviations used in Figure 1, esp. “Ac“ and “Me“ need tob e explained in the legend.

Author Response

  1. A huge, I would even say impressive, work is hidden behind this manuscript, especially Table 1. However:

1A. Was the literature search systematic? If not, this should be mentioned as a limitation as some important findings might have been omitted.

  • We searched the literature in pubmed database by looking for mechanisms of action of a given compound and considered only publications not older than 4 years. Unfortunately, we cannot call our literature searching systematic, as we did not document our searches during the process, used one database only, and did not use any elaborated approach to term finding. Appropriate note was placed in the text.

1B. I would divide Table 1 into several smaller subtables, e.g. with classical epigenetic mechanisms, miRNA, transcription factors.

  • We divided the Table into 4 ones.

1C. The content of this table needs to be, at least concisely, discussed in the main text of the manuscript, the best in sections corresponding to the sub-tables mentioned above.

  • As the review is focused on the influences of dietary compounds on (epi)genetics, we believe that the part regarding the influence of ingested compounds on disease development should be narrowed to one chapter of the manuscript, and, where necessary, references to the Tables were added in the text. However, if the Reviewer finds it necessary to add more information on the diseases/dietary components interactions, we will gladly introduce such changes.

  1. In continuation, some topics must be naturally ignored as not all possible aspects can be covered. But I would definitely not omit allergies. There are reports on the role of food compounds such as fish or olive oil-deriving compounds such as PUFA or polyphenols in regulating the mechanisms underlying the development of allergies (PMID: 28159873 and 30823645). Or milk components such as fat-soluble vitamins A and D, as well as the water-soluble B vitamins, and PUFA (PMID: 28933365, 33193294). And others. Those should be mentioned.

  • We added more data on allergies (in terms of (epi)genetic mechanisms of the dietary compounds).

  1. The Authors discuss indirect effects of dietary compunds on gene expression. But there are also direct effects exerted by regulatory RNAs present in foods such as:

3A. milk (PMID: 26707195, 34067156).

3B. and others (PMID: 33567805, 33668787).

Those should be mentioned, at least shortly.

  • We amended the text of the manuscript as suggested

  1. I guess, SCFA as indirect dietary compunds (gut bacteria oligosaccharide fermantation metabolites) can be ignored in this work (as they currently are).
  2. Please, check the list of abbreviations. Some of them are never used in the manuscript, e.g. PUFA.

  • We checked and corrected the list of abbreviations

  1. Abbreviations used in Figure 1, esp. “Ac“ and “Me“ need tob e explained in the legend.

  • We added explanation to these abbreviations in Figure 1 caption.

Round 2

Reviewer 2 Report

My comments have been addressed well. I have no further reservations.

Author Response

Thank you.